# A Novel Traveling-Wave-Based Method Improved by Unsupervised Learning for Fault Location of Power Cables via Sheath Current Monitoring

**DOI:** 10.3390/s19092083

**Published:** 2019-05-05

**Authors:** Mingzhen Li, Jianming Liu, Tao Zhu, Wenjun Zhou, Chengke Zhou

**Affiliations:** 1School of Electrical Engineering and Automation, Wuhan University, No. 299, Bayi Road, Wuchang District, Wuhan 430072, China; 2Chinese Society for Electrical Engineering, No.1, Baiguang Road, Xuanwu District, Beijing 100761, China; 3State Grid Hubei Electric Power Company, State Grid Corporation of China, No. 1701 Jiefang Avenue, Jiang’an District, Wuhan 430013, China; zt1231036@yeah.net; 4School of Engineering and Built Environment, Glasgow Caledonian University, Cowcaddens Road, Glasgow G4 0BA, UK

**Keywords:** circuit faults, fault currents, fault location, power cables, sheath currents

## Abstract

In order to improve the practice in maintenance of power cables, this paper proposes a novel traveling-wave-based fault location method improved by unsupervised learning. The improvement mainly lies in the identification of the arrival time of the traveling wave. The proposed approach consists of four steps: (1) The traveling wave associated with the sheath currents of the cables are grouped in a matrix; (2) the use of dimensionality reduction by t-SNE (t-distributed Stochastic Neighbor Embedding) to reconstruct the matrix features in a low dimension; (3) application of the DBSCAN (density-based spatial clustering of applications with noise) clustering to cluster the sample points by the closeness of the sample distribution; (4) the arrival time of the traveling wave can be identified by searching for the maximum slope point of the non-noise cluster with the fewest samples. Simulations and calculations have been carried out for both HV (high voltage) and MV (medium voltage) cables. Results indicate that the arrival time of the traveling wave can be identified for both HV cables and MV cables with/without noise, and the method is suitable with few random time errors of the recorded data. A lab-based experiment was carried out to validate the proposed method and helped to prove the effectiveness of the clustering and the fault location.

## 1. Introduction

In order to shorten the repair time of faulted electrical transmission lines, efforts have been made to develop reliable and accurate fault location methods [1,2,3]. Most of them are applied for overhead lines: Fault location methods for power cable designs are rarely reported. Among the fault location methods for transmission lines, the impedance-based method and the traveling-wave-based method are the two most popular methods [4,5,6]. Actual operating experience indicates that additional effort is still needed to locate short circuit faults in power cable systems more accurately.

The structure of cable lines and the laying environment are more complicated than those of overhead lines, which makes fault location more difficult. A practical power cable circuit may contain a few major sections of underground cable and/or overhead lines, which makes it difficult to determine accurately the relationship of impedance versus distance when applying an impedance-based method. In addition, accurate electrical parameters of each power cable are not always available. Furthermore, the multilayer metal structure and the close line distance can cause serious attenuation and distortion of traveling waves in cable systems, which makes the noise elimination and the accurate identification of the wave head more difficult.

Recently, there were reports of several kinds of improvements to the impedance-based and traveling-wave-based methods. In References [7] and [8], the impedance-based method was improved for use in double circuit, medium power distribution networks. The method takes into account the mutual inductive effect of double circuit lines, and it is applicable to fault location in scenarios of double circuit lines. However, the accuracy of the line parameters is essential to the method. In References [9] and [10], improvements were made to the traveling-wave-based method, i.e., using the time difference between the first incident traveling wave and its first reflection wave to avoid synchronization problems with multiple signals. Also, in Reference [10], a normalized fault location formula, which requires neither an external common time reference nor the traveling wave velocity, was proposed. Despite these advantages, the signal time differences used in Reference [10] can be confusing if the fault contains multiple discharges, which is common in power cable faults.

As a result, effort is still needed to tackle the noise and time deviation problems, which has always been an important factor. In the past 20 years, machine learning methods (accompanied with the image processing) has undergone rapid developments [11]. As one of the machine learning methods, the unsupervised learning has been widely used in feature reduction and clustering [11]. In Reference [12], an unsupervised-learning-based approach was proposed for automated defect inspection on textured surfaces, and convolutional denoising autoencoder network was introduced to reconstruct image patches. In Reference [13], an unsupervised learning technique was proposed to optimize radio maps for indoor localization, a route mapping filter was designed based on the Viterbi path, a technique related to hidden Markov models. In Reference [14], an unsupervised manifold learning was used to study the time evolution of the state of a continuously monitored quantum system. All these researches contributed greatly to the application of the unsupervised learning methods to real world measurements. 

For the first time, this paper introduces an unsupervised learning method for fault location of power cables and to improve on the traditionally applied traveling-wave-based method. The improvement mainly lies in the identification of the arrival time of the traveling wave. Instead of the load currents (core currents), the sheath currents were monitored for the traveling wave. This can be achieved in a more convenient manner and the characteristics of the traveling wave can be intensified by the matrix constituted by the sheath currents of the cables. The arrival time of the traveling wave can be identified via t-SNE dimensionality reduction and DBSCAN clustering. Simulations and calculations were carried out for both HV (high voltage) and MV (medium voltage) cables. Results indicate that the arrival time of the traveling wave can be identified for both HV cables and MV cables when noise is absent or it is present. The method has the advantage of few random time errors in the recorded data. Finally, the effectiveness of the fault location method has been evaluated by a short-circuit fault test on a cable circuit.

## 2. Theoretical Basis of Traveling-Wave Method

The accuracy of the traveling-wave method is dependent on the accuracy of determination of the propagation time of the transient traveling wave on the faulty line between the line terminal and the fault point. The basic principle of the traveling-wave method [15] is shown in Equation (1), where *x* represents the distance from the fault point to the line terminal detection point, *v* represents the propagation velocity of the traveling wave, and *t* represents the propagation time of the traveling wave.
(1)x=vt

Based on Equation (1), ideally, the traveling-wave process can be shown in Figure 1, where *M* and *N* represent the two ends of the transmission, or power, line; *L* represents the length of the power line; *F* represents the location of the fault; *t_M_* represents the time at which the fault traveling wave reaches *M*; *t_N_* represents the time at which the fault traveling wave reaches *N*; *t_Mr_* represents the time when the first reflection wave, i.e., the wave reflected from the fault site, reaches *M*; *t_Nr_* represents the time when the first reflection wave reaches *N*; *t_Mt_* represents the time when the first refraction, i.e., the wave refracted from the fault site, wave reaches *M*; *t_Nt_* represents the time when the first refraction wave reaches *N*.

As the fault start time cannot be detected directly and accurately in practice, the propagation time of the traveling wave *t* in Equation (1) is unknown. Therefore, classical single-terminal traveling-wave method estimates the fault position using Equation (2) instead, where *D_MF_* represents the length of the section of the line between *M* and the fault *F*.
(2)$$D_{MF}=\frac{1}{2}v\left(t_{Mr}-t_{M}\right)$$

Similarly, if the observation point is *N*, the fault location can be estimated by Equation (3), where *D_NF_* represents the length of the section of the line between *N* and the fault *F*.
(3)$$D_{NF}=\frac{1}{2}v\left(t_{Nr}-t_{N}\right)$$

However, there will be an error in determining the fault location if *t_Mr_* is confused with *t_Mt_* the arrival time of the wave reflected from the termination beyond the fault position. If the power line contains several sections of power cable and overhead line, the multiplicity of traveling wave reflections are more complicated. The two-terminal traveling-wave method is developed for the situation with the two ends monitored. Equations (4) and (5) present the basic theory of the two-terminal traveling-wave method.
(4)$$\left\{ \begin{array}{l} \frac{D_{MF}}{v}-\frac{D_{NF}}{v}=t_{M}-t_{N}\\ D_{MF}+D_{NF}=L \end{array}\right.$$
(5)$$\left\{ \begin{array}{l} D_{MF}=\frac{1}{2}\left[v\left(t_{M}-t_{N}\right)+L\right]\\ D_{NF}=\frac{1}{2}\left[v\left(t_{N}-t_{M}\right)+L\right] \end{array}\right.$$

As shown in Equations (4) and (5), the two-terminal traveling-wave method requires only the identification of the arrival times of the first traveling waves (*t_M_* and *t_N_*), and no reflected wave times are required. However, the two-terminal traveling-wave method is dependent on accurate calculation of (*t_M_* − *t_N_*), i.e., the clocks of the local and the remote sensor must be consistently synchronized. Any loss of the synchronicity of the time stamp may cause the failure of the method.

## 3. Typical Cable Structures and the Monitoring of the Fault Signals

### 3.1. Typical Cable Structures

Typical power cables for high voltage (HV: Above 110 kV) have a single-core structure and for medium voltage (MV: 10~35 kV) a three-core structure, as shown in Figure 2a,b, respectively.

Overhead lines rely on air insulation, power cable lines have multiple layers of solid insulation and metal structures, which makes the calculation of line parameters more complicated. Considering Figure 2a, since the metal sheath of the single-core cable has at least one end grounded, the potential of the metal sheath is close to the ground potential during normal operation. In addition, as the load current flows through the cable core, the direction of the electric field on the cable cross section is radial, shown as ***E*** in Figure 2a, and the direction of the magnetic field is shown as ***H***. Therefore, the direction of the Poynting vector of the energy transmission is the ***E*** × ***H*** direction, which means the energy propagates along the direction of the main insulation between the two layers of metal. When a breakdown occurs at any position of the cable, there are fault signals on both the core conductor and the metal sheath. Therefore, the arrival time of the fault signal for the core conductor or the metal sheath is the same. For safety and convenience, this paper will obtain the fault signal by monitoring the sheath current. For the three-core cable structure shown in Figure 2b, the metal sheath and the metal armor both need to be grounded at the same grounding position. Therefore, the monitoring of the sheath current can also obtain the arrival time of the fault signal for three-core cables.

There are at least two kinds of breakdown process [16] for power cables. First, the main insulation fails in an instant, and there are no signs (such as partial discharge) before the breakdown. Second, the main insulation is gradually deteriorating, in which case multiple discharges may occur before a breakdown channel is formed, i.e., the deterioration process of the insulation is accompanied and exacerbated by partial discharge. For the second breakdown process, the first reflection wave from the fault position may be confused with the traveling wave from the second/third/fourth… discharge, which makes the single-terminal traveling-wave-based method invalid. 

### 3.2. The Monitoring of the Fault Signals

The sheath current monitoring system is presented in Figure 3, where a typical cross-bonded HV cable is shown as an example. The sheath current monitoring system contains four parts, namely, the data acquisition module, the communication module, the location analysis software installed in the cloud server, and the interface for the final users, e.g., cable maintenance engineers. The data acquisition module (including the sheath current sensors) are employed at the grounding boxes and the cross-bonding link boxes. When a fault happens in a cable section, the sheath current in the loop where the fault happens will rise to the level of fault current. Meanwhile, the data acquisition module will be woken up and start data uploading. The communication module of the data acquisition system can transmit the recorded data to a designated cloud server, where the location analysis software carries out data analysis before sending the location results to maintenance engineers. The recoded data can also be downloaded from the server for further analysis.

## 4. Analysis of the Sheath Currents and the Fault Location using Unsupervised Learning 

This paper proposes an autonomous learning mechanism based on the two-terminal traveling-wave method, which enables the algorithm to accurately identify the accurate arrival time of a fault traveling wave with multiple monitoring data. The fault location errors of the two-terminal traveling-wave method mainly come from the identification of the arrival times at the two ends of the line, and the main sources of the identification errors are randomness, such as the time delays of the synchronization, the noise interference, etc. In order to reduce the random error, the sheath currents of three-phase HV cables and the adjacent cables of the same cable passage are monitored, so that a multidimensional space matrix can be built, and more accurate arrival times could be identified from analysis of the multidimensional matrix. The process of identifying the arrival times at the two ends of the line is shown in Figure 4.

As shown in Figure 4, a high-dimensional space matrix needs to be built from information of all the monitored sheath currents, then the high-dimensional matrix can be mapped into a two-dimensional space by t-SNE (t-distributed Stochastic Neighbor Embedding) [17], which is a data visualization technique used for dimensionality reduction based on data characteristics. It is to be noted that the sheath current matrix built in step one can be two-dimensional or higher, and more data and higher dimensional matrix is recommended for the less randomness of the data. The t-SNE process not only reduces the data dimension, but also reduces the possible time deviation in the original data. The two-dimensional data is then clustered using DBSCAN, which is a density-based clustering algorithm. The possible noise in the data can be identified during the DBSCAN processing. Finally, the arrival times of the traveling wave can be identified using the cluster data. The specific four steps of the process are demonstrated in Section 4.1, Section 4.2, Section 4.3 and Section 4.4, with a simulation case.

### 4.1. The Construction of the Sheath Current Matrix

As stated, for safety and convenience, instead of monitoring load currents (core currents), the sheath currents were monitored for the traveling wave from faults. Thereafter, the characteristics of the traveling wave can be intensified from the matrix constituted by the sheath currents of the cables. In practical HV transmission systems involving cross-bonded cables, the sheath currents can be monitored at cable terminals and cross-bonding boxes (HV cables). Usually, in such a case, there are multiple cables sharing the same passage/duct. If a short-circuit fault occurs in a cable, inductively coupled signals can be detected on the sheaths of adjacent cables. The accuracy of the determination of arrival time of the traveling wave, through correct identification of signals, is believed to be improved if multiple sheath currents are analyzed together. Hence, a high- dimensional matrix can be built using the amplitude of each sheath current as the matrix row vector, and the number of sheath currents being measured is the number of columns of the matrix.

For a more intuitive demonstration, a short 110 kV HV cable circuit (500 m) model was established using PSCAD (Power System Computer Aided Design). The short cable circuit uses the single-point bonding technique, i.e., there is only one direct earthing point for the metal sheath as shown in Figure 5. An additional connection to ground is through the sheath protector, a normally high impedance path. The short cable circuit is buried directly in the ground with horizontal laying, shown in Figure 6. The parameters of cross-sectional structure of the cable are shown in Table 1.

A short-circuit fault has been set at 400-m from cable terminal 1 of phase A cable, as shown in Figure 5. The sheath currents of three phases are measured at terminal 1 and have been analyzed with the unsupervised learning method in this case. The sheath currents of the cable are shown in Figure 7, the distortion due to the arrival of a traveling wave is seen in *I*_1*a*_.

As shown in Figure 7, the sheath current of the faulty phase is easy to distinguish by the traveling wave. The three sheath current waveforms can be used to construct a three-dimensional matrix *U* ∈ ***R****^N^*^×3^, where *N* and *d* represent the number and dimension of samples, as plotted in Figure 8 for this case. In the simulation the sampling rate is 100 MHz and the sampling time is 12 μs, therefore *N* = 100 × 10^6^ × 12 × 10^−6^ = 1200. The matrix *U* is the input of the t-SNE algorithm. 

### 4.2. Dimensionality Reduction by t-SNE

The t-SNE (t-distributed Stochastic Neighbor Embedding) algorithm is a nonlinear dimensionality reduction technique for embedding high-dimensional data for visualization in a low-dimensional space of two or three dimensions. Unlike traditional linear dimensionality reduction techniques, which focus on global structures, or traditional nonlinear techniques, which prefer to maintain local structures, t-SNE can reveal both global and local structures in high-dimensional data simultaneously [17]. Therefore, t-SNE is a good choice for unsupervised machine learning.

For the high-dimensional matrix *U* ∈ ***R****^N^*^×*d*^ constructed of the sheath current data, *N* represents the sum number of the data of multiple monitoring points, *d* represent the number of sampling points. The t-SNE algorithm aims to map the high-dimensional matrix to a low dimensional matrix *V* ∈ ***R****^N^*^×*a*^, where *a* equals 2 or 3. For convenience, *a* is set to 2 in the subsequent analysis.

The dimensionality reduction is accomplished by an optimization process with the aim shown in Equation (6), where *p_ij_* represents the similarity between the *i*-th column vector *u_i_* of the matrix *U* and the *j*-th column vector *u_j_*, and *q_ij_* represents the similarity between the *i*-th column vector *v_i_* of the matrix *V* and the *j*-th column vector *v_j_*.
(6)$$\min\;\;C=\sum_{i}\sum_{j}p_{ij}\log\frac{p_{ij}}{q_{ij}}$$

The restrictions of the optimization process are defined by *p_ij_* and *q_ij_*, whose equations are represented in Equations (7)–(9).
(7)$$p_{ij}=\frac{p_{j|i}+p_{i|j}}{2N}$$
(8)$$p_{j|i}=\frac{\exp\left(-{\left\|u_{i}-u_{j}\right\|}^{2}/2\sigma_{i}^{2}\right)}{\sum_{k\neq i}\exp\left(-{\left\|u_{i}-u_{k}\right\|}^{2}/2\sigma_{i}^{2}\right)}$$
(9)$$q_{ij}=\frac{\exp{\left(1+{\left\|v_{i}-v_{j}\right\|}^{2}\right)}^{-1}}{\sum_{k\neq l}\exp{\left(1+{\left\|v_{k}-v_{l}\right\|}^{2}\right)}^{-1}}$$

In Equation (8), *p_j|i_* represents the conditional probability of the data *u_j_* being similar to data *u_i_*, which means *u_i_* would pick *u_j_* as its neighbor if neighbors were picked in proportion to their probability density under a Gaussian centered at *u_i_*.

There are three main steps of the optimization process: (1) Convert the high-dimensional Euclidean distance between data points into a conditional probability of similarity *p_ij_*; (2) calculate the conditional probability of similarity of the points of the low-dimensional matrix *q_ij_*; (3) map the high-dimensional data into a two-dimensional space by minimizing the mismatch between *p_ij_* and *q_ij_*. More simply, the dimensionality reduction process can be analogy to the shadow of a three-dimensional object. Imagine there is a beam of light that is directed at the three-dimensional scatter (e.g., Figure 7), the shadow of the three-dimensional scatter is a two-dimensional scatter. Two points (or multiple points) in space in the direction of the beam may coincide with one point on the shadow. The process of the dimensionality reduction is similar to the process of overlapping these spatial points as much as possible on the shadow.

By applying t-SNE to a high-dimensional matrix *U*, a two-dimensional matrix *V* can be obtained, such as that plotted in Figure 9. 

As shown in Figure 9, the scatter diagram of the two-dimensional map is more concentrated than the three-dimensional map shown in Figure 8: There is no obvious noise in either figure. It is to be noted that there are no legends on the axes of the two-dimensional map after t-SNE, for there are only mathematical meanings of the axes. The dots in Figure 9 are the dots of the original sheath current matrix optimized by t-SNE (similar to the “overlapping” process). The matrix *V* is the optimization result of the t-SNE algorithm, which means the matrix *V* is in closest agreement with Equations (6)–(9). Unfortunately, the physical meaning of the horizontal and vertical coordinates of the matrix *V* no longer exists after the t-SNE process. The matrix *V* is the input of DBSCAN (density-based spatial clustering of applications with noise) clustering.

### 4.3. DBSCAN Clustering

DBSCAN [18] is a density-based clustering algorithm. Such density clustering algorithms generally assume that categories can be determined by the closeness of the sample distribution. DBSCAN uses parameters *Eps* and *MinPts* to describe the closeness of the sample distribution in the neighborhood: *Eps* describes the neighborhood distance threshold of a certain sample and *MinPts* describes the threshold of the number of samples in the neighborhood of a sample with a distance of *Eps*. 

There are four main steps within DBSCAN: Find a data point *p* that has not been visited in the dataset *D*, mark *p* as visited, then search the neighbors of *p* within a radius of *Eps*, set the neighbor data as *Ne*. If the data number of *Ne* is less than *MinPts*, mark all the data of *Ne* as noise. Otherwise, mark all the data of *Ne* as a new cluster *C* (*C* = {*C*_1_, *C*_2_, …, *C_k_*}).Expand clusters *Ne* and *C*: Find a data point *q* that has not been visited in the dataset *Ne*, mark *q* as visited, then search the neighbors of *q* within a radius of *Eps*, set the neighbor data as *NE*. If the data number of *Ne* is more than *MinPts*, combine *Ne* and *NE*, denoted as *Ne*. If *q* is not member of any clusters, add *q* to *C*.Repeat step (2) until *Ne* is empty.Repeat steps (1)–(3) until every point in *D* has been visited and been marked as a member of a cluster or as noise.

Each cluster *C* can be output by applying DBSCAN to the two-dimensional matrix *V* with parameters *Eps* and *MinPts*.

According to the two-dimensional map of Figure 9, there are multiple visible clusters: The distances between clusters are significant, the distances between the points of each cluster are small. For effective DBSCAN clustering in this example, *Eps* and *MinPts* are set to 0.1 and 7, respectively. The results of the clustering process, which resulted in fourteen clusters being identified, can be seen in Figure 10. As the sheath currents are sampled at equal intervals, the crests of the traveling waves are the signal with the maximum amplitude and with the fewest samples during the time interval. Once a fault happens to a cable (phase), the wave crests will also be induced in the sheath current of the adjacent cables (phases). Thus, the waveforms of the sheath currents are very similar, and these sampling points of the sheath currents are relatively concentrated in the distribution of the two-dimensional map after t-SNE. 

As shown in Figure 10, there are 14 clusters with no indication of noise. The clustering results are consistent with the simulation without added noise, and the clusters were distinguished from each other based on the closeness of the sample distribution.

### 4.4. The Identification of the Arrival Time

For further analysis, the number of samples in each of the 14 clusters is presented in Figure 11.

As shown in Figure 11, clusters 2 and 4 have the largest number of samples, and clusters 1 and 6 have the lowest. As the crest of the traveling wave is the signal with the maximum amplitude, the cluster of the crest should be the one with the fewest samples. Therefore, the subsequent algorithm steps are as follows:Identify each sample of the cluster(s) with the lowest number of samples (cluster 1 and cluster 6 in this case).Match the samples with the original sheath current samples of the faulty cable and retrieve the data for each sample.Find the maximum slope points of each cluster based on the original sheath current sample data.Output the corresponding time of the selected cluster data.

The time of the maximum slope point corresponds with the arrival time of the traveling wave. In this case, the corresponding times are 2.67 ms and 5.33 ms: These times are consistent with the arrival time of the first traveling wave and the second traveling wave, as shown in Figure 12.

In Figure 12, the waveform has been recorded at the very time when the fault just occurred. As the fault simulated is of a penetrating breakdown, the sheath current waveform continues to grow during the first 12 μs. The regular propagation pattern of the electromagnetic wave was presented in Figure 6, with the first traveling wave arriving at 2.67 μs and the second arriving at 5.33 μs. As the traveling wave velocity is determined by the cable parameters, it can be calculated using Equation (10), where *μ* is the magnetic permeability and *ε* is the insulation permittivity.
(10)$$v_{0}=\frac{1}{\sqrt{\mu \varepsilon }}$$

In this case, *μ* = 4 × 10^−7^ and *ε* = 3.63 × 10^−11^ and so *v*_0_ = 1.48 × 10^8^ m/s. Based on this, the fault position can be located using Equation (2), i.e., the single-terminal traveling-wave method. It can be determined that the fault is located at 400 m away from cable terminal 1, which is consistent with the initial data for the simulation.

## 5. The Application of the Unsupervised Learning Method with Typical Cable Structures

The case study presented in Section 4 demonstrated the application of the unsupervised learning method for a short, single-bonded HV cable circuit. For HV cable circuits there are more complicated bonding techniques, double-bonded, middle-bonded, cross-bonding, etc. Three-core MV cable has another bonding method. However, the application of the proposed method is quite similar and the results of investigation of the different bonding techniques will be presented in this section.

### 5.1. Application to a Cross-Bonded HV Cable Circuit

First, a model of a 1500 m cross-bonded 110 kV HV cable circuit was established using PSCAD. The cable parameters are the same as those presented in Table 1, while the schematic diagram of the cable circuit and the arrangement of the current sensors are presented in Figure 13. As shown, the major section of the cable circuit is 1500 m, and the major section is constituted by three minor sections, each 500 m long. The current sensors are located at the grounding boxes, G1 and G2, and the cross-bonding link boxes, J1 and J2. For this study, the cross-bonded cable circuit is also buried directly in the ground with horizontal laying, as shown in Figure 6.

A short-circuit fault has been set within cable section A1, 300-m from G1. The sheath currents measured at the 12 sensors in the cross-bonded cable circuit are used to form the 12-dimensional matrix *U*, which is the input of the t-SNE algorithm. The 12 sheath currents generated in the simulation are presented in Figure 14. As in the previous case study, *Eps* and *MinPts* are set to 0.1 and 7 for DBSCAN clustering. Based on this, the clustering results are as shown in Figure 15, and the number of the sheath current samples in each cluster is shown in Figure 16.

As shown in Figure 15 and Figure 16, there are 13 clusters: clusters 6 and 5 have the lowest number of samples. In this case, cluster 5 has a time for the maximum slope point of sheath current *I*_1*a*_ of 2.03 μs, in cluster 6 the corresponding time of sheath current *I*_2*a*_ is 1.35 μs. These are consistent with the expected times of the first traveling wave. 

### 5.2. Application to a MV Cable Circuit

A typical MV cable structure was shown in Figure 2. The metal sheath and the metal armor are connected and grounded at the two ends, as shown in Figure 17: Note that phase A, B, and C are all represented by the single line “core conductor”. For this case study, the MV cable circuit is 400 m long and the short-circuit fault was set in phase A, 200 m from cable terminal 1. The structure parameters of the cable are presented in Table 2, and the material parameters of the main insulation (XLPE) are the same as those in the HV cables. The sheath current waves of monitored by the two current sensors, sited as in Figure 18, in further analyses *I_m_*_1_ is the sheath current recorded by current sensor 1 and *I_m_*_2_ is the sheath current recorded by current sensor 2.

As there are only two sheath currents, the dimensionality reduction process for t-SNE will not be necessary. The sheath current matrix will be used for DBSCAN clustering directly. The clustering results are shown in Figure 19, and the number of samples in each cluster is shown in Figure 20.

Clearly, cluster 1 has the fewest samples and, from examination of Figure 20, the sample distribution is very concentrated, i.e., as discussed, related to the difference in arrival times at the terminals. To illustrate the concentrated distribution of the samples from cluster 1, the scatter diagram of the sheath current *I_m_*_2_ is replotted in Figure 21, where the corresponding samples of the cluster 1 are highlighted. 

As shown in Figure 21, the samples of cluster 1 are almost coincident with the values of the first traveling waveform; the amplitude of these samples are closest to the corresponding samples of *I_m_*_1_. The arrival time can be identified as 1.35 s by the same method, which is also consistent with arrival time of the first traveling wave.

### 5.3. A Fault Location Test on a MV Cable Circuit

In order to test the effectiveness of the fault location method in practice, a fault location test was carried out on a MV cable circuit. The test cable platform, constructed from four reels of three-core XLPE cable (YJV22-8.7/15kV-3*240), was built at the Wuhan UHV (Ultra High Voltage) Test Base. All the cable sheaths have been grounded together at the terminal. The four reels of cable are connected in series to form a test circuit, shown schematically in Figure 22. The middle two sections (about 300 m each) constitute the monitored circuit. The middle two sections are connected by a manually constructed defective joint, and the six cable ends are connected by 10 kV cable outdoor terminals. On-site photographs are presented in Figure 23.

The test was carried out on a sunny winter day (24 January 2019). First, the manually constructed defective joint was connected in the cable and the current sensors were installed in the test circuit, as shown in Figure 23. A series resonant source is used to raise the voltage step-by-step until a breakdown occurs in the defective cable joint. During the process of raising the voltage there were several obvious discharges, visually and/or audibly observed, before the penetrating breakdown occurs. As the data sampling rate is high (50 MHz), the sampling time cannot be too long (dozens of microseconds). Normally, the first recorded data is used for fault location. However, as the buffer of the data recording device is not designed for long time sampling, i.e., the length of time to raise the voltage step-by-step, incorrect operation or continuous discharge may rewrite the buffer with data from subsequent discharge events. To illustrate the superiority of the fault location method, the discharge waveforms of the subsequent data have been recorded. The recorded discharge waves, using normalized amplitudes, are presented in Figure 24: *I_L_* and *I_R_* are the currents recorded by the sensor to the left and right hand of the fault, respectively.

As shown in Figure 24, although the subsequent waveforms slightly deviate, the first crest of the recorded discharge signal waveforms of *I_L_* and *I_R_* are substantially coincident. This indicates that the time of arrival of the traveling waves at the monitoring positions is basically the same, and the fault is located at the midpoint of the electrical distance between the two monitoring positions. The DBSCAN clustering results of *I_L_* and *I_R_* are presented in Figure 25, where points labeled as -1 represent noise, and those labeled 1~3 represent three types of data points, respectively. The scatter diagram of the clustering data of *I_L_* and *I_R_* is replotted in Figure 26.

In Figure 26, A1 and A2 are the points of cluster 1 which correspond to *I_L_* and *I_R_* respectively: These points are completely coincident. B1 and B2 are the points of cluster 2 which correspond to *I_L_* and *I_R_*, respectively. Although the differences in these points is small, they are not completely coincident. C1 and C2 are the points of cluster 3 which correspond to *I_L_* and *I_R_*, respectively. These points are closer together. D1 and D2 are the points of cluster −1 which correspond to *I_L_* and *I_R_*, respectively. These points are far apart. According to the analysis, the first crest of the traveling waves of *I_L_* and *I_R_* are completely coincident, and the time difference is 0, so that the fault point can be identified as the midpoint of the cable section being examined. In cases where the length of the cable is known, e.g., distance between the current sensors is 297 + 298 m, the fault location result is 297.5 m away from the monitoring position, which is 0.5 m from the actual deviation.

As the traveling waves from the mid-position arrive at the same time, the signals from discharges before or after the breakdown can also be used for fault location. The validity of the clustering process and fault location in this case study demonstrate the effectiveness of the method.

## 6. The Comparison of Several Popular Algorithms for the Arrival Time Identification

The superiority of the fault location method of the paper mainly lies in the processing of multiple signals, especially when there is time deviation in a few signals as a result of external noise or signal digitization. To illustrate the superiority of the presented method, several popular algorithms for identification of signal arrival time have been compared. They are: The threshold method [19,20]: Searches for the first crest of the traveling signal with a set threshold and uses the difference of the trigger time detected by the threshold on the rising edge (or falling edge) of the first crest as the time difference of the two signals. Clearly, this method is susceptible to noise, and a single threshold cannot be determined for all cases. An improvement to this would have the threshold being set by assessment of the first traveling wave crest, with the threshold value being used for subsequent data capture.The wavelet-based multi-scale time-frequency analysis method [21,22,23]: The arrival time is identified by the maximum position of the wave crest signal. This method is mainly used for overhead lines, as they have a simpler structure and the method is very susceptible to errors in the accuracy of the line parameters. Ideally, line parameters should be accurate for location of the fault.The cross-correlation algorithm [24,25]: The time difference between two signals is determined by the maximum of the cross-correlation function. This method is mainly used for situations where the propagation path is a homogeneous medium, and there is minimal noise. The cumulative energy method [26,27]: The cumulative energy of the signal is determined by the sum of the squares of each sample, with or without a correction term of negative slope. The arrival time of the wave is determined by the absolute minimum of the cumulative energy. Again, the problem of this method still lies in the noise, especially for the slope of the noise wave. The method presented in this paper: t-SNE+DBSCAN. The superiority of the method of this paper mainly lies in the processing of multiple signals. However, as this method uses highly nonlinear algorithms, the calculation time is longer.

Simulations on MV cables, in which two signals are assessed, and cross-bonded HV cables, in which 12 signals are assessed, were carried out using the five algorithms. 

First, for the MV cable as shown in Figure 17, the fault position was set, in turn, at 50 m, 100 m, 150 m, 200 m, 250 m, 300 m, and 350 m from current sensor 1. The corresponding time differences are 2.03 μs, 1.35 μs, 0.68 μs, 0 μs, −0.68 μs, −1.35 μs, and −2.03 μs, respectively. The time difference results for the five algorithms are presented in Table 3.

As shown in Table 3, the time difference can be identified accurately by all algorithms under ideal conditions in MV cables. 

Similar simulations and calculations were carried out for the cross-bonded HV cable circuit, shown in Figure 13. To make the investigation more realistic, the fault position in minor cable section A1 is 300 m from the left monitoring position (G1) and white noise, between 0~50 dB, was added to the waveforms. The time difference between the traveling waves reaching detection points G1 and J1 should be −0.68 s: Results from each algorithm are presented in Table 4.

As shown in Table 4, the five algorithms have different sensitivities to noise. Methods A~E have, sequentially, better ability to reject noise. When the noise level is too high, all five algorithms lose accuracy in different degrees. In practice, there are long cable lines which have a minimal number of sensors, whose recorded noise is too high or whose synchronization accuracy is not enough to allow accurate location of faults. In order to reduce the effect of randomly generated errors, multiple recorded data from a power line can be analyzed together using method E. 

Additionally, still with the cross-bonded HV cable circuit shown in Figure 13 and with the fault position in A1 being 300 m from the left monitoring position (G1), a random error was added to the arrival time at the current sensor *I*_1*a*_, e.g., simulating loss of the synchronization timestamp. The tolerance range of the random error applied is [−1 μs, +1 μs]. The performances of the five algorithms under this condition are presented in Figure 27.

As shown in Figure 27, when there is a time deviation within the recorded data from *I*_1*a*_, the performance of method E is much better than those of methods A~D, which must improve the performance for fault location. Although this issue has a low probability of occurring in practice, when it does occur, it is catastrophic for fault location when using other algorithms. The results indicate that method E is suitable for use in fault location with a few random time errors in the recorded data.

## 7. Discussions and Conclusions 

### 7.1. Discussions

Similar to most learning algorithms, the method has certain limitations. As shown in Table 4, the method loses accuracy when noise level is too high, the critical noise level for the effectiveness of the method deserves further research. The method of reducing the time deviation using multiple signals is not used in the case where the time deviation is highly dispersed. If there are time deviations among all the recorded data and the average errors are not negligible, the fault location errors of the method could be high. In order to locate the fault accurately, all minor sections of the cable are recommended to be equipped with the current sensors at current stage. The feasibility of reducing the number of the current sensors will be studied in the future.

### 7.2. Conclusions

This paper proposes a novel traveling-wave-based method for fault location in power cables. The proposed approach consists of four steps: (1) The traveling wave associated with the sheath currents of the cables are grouped in a matrix; (2) the use of dimensionality reduction by t-SNE (t-distributed Stochastic Neighbor Embedding) to reconstruct the matrix features in a low dimension; (3) application of the DBSCAN (density-based spatial clustering of applications with noise) clustering method to cluster the sample points by the closeness of the sample distribution; (4) the arrival time of the traveling wave can be identified by searching for the maximum slope point of the non-noise cluster with the fewest samples. Simulations and a fault location test have been carried out to successfully validate this method. From the results it can be concluded that:(1)The arrival times of traveling waves can be identified accurately by the proposed method.(2)The application of the algorithm helped to improve the accuracy of the traveling-wave-based fault location method.(3)The proposed method can be applied to both HV cables and MV cables (the sheath current monitoring positions are different).(4)The superiority of the proposed method mainly lies in the processing of multiple signals, especially when there is time deviation of a few signals.(5)Five popular algorithms for determining wave arrival times were compared. Results indicate that, although the arrival time can be identified accurately by all the five algorithms under ideal conditions, the performance of the proposed method is better than the other four algorithms when white noise is present in the waveforms.(6)The performance of the proposed method is better than the other four algorithms when there is a random time deviation within the recorded data.

## Figures and Tables

**Figure 1 sensors-19-02083-f001:**
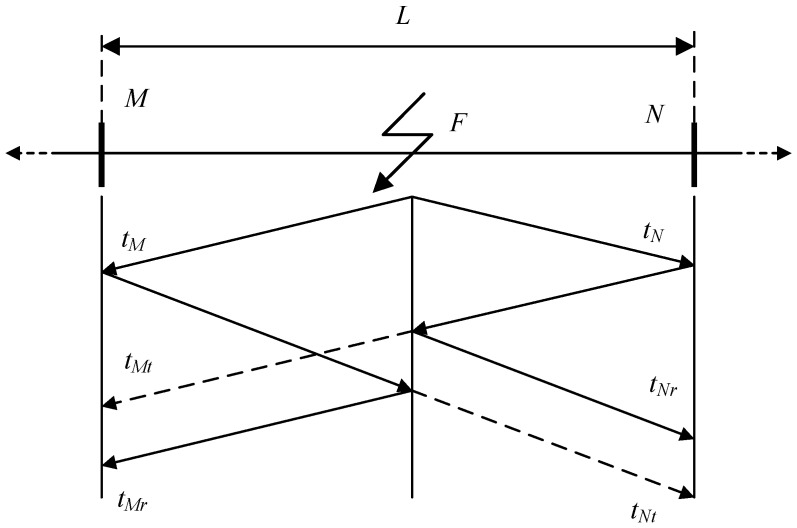
The schematic diagram of the traveling wave process.

**Figure 2 sensors-19-02083-f002:**
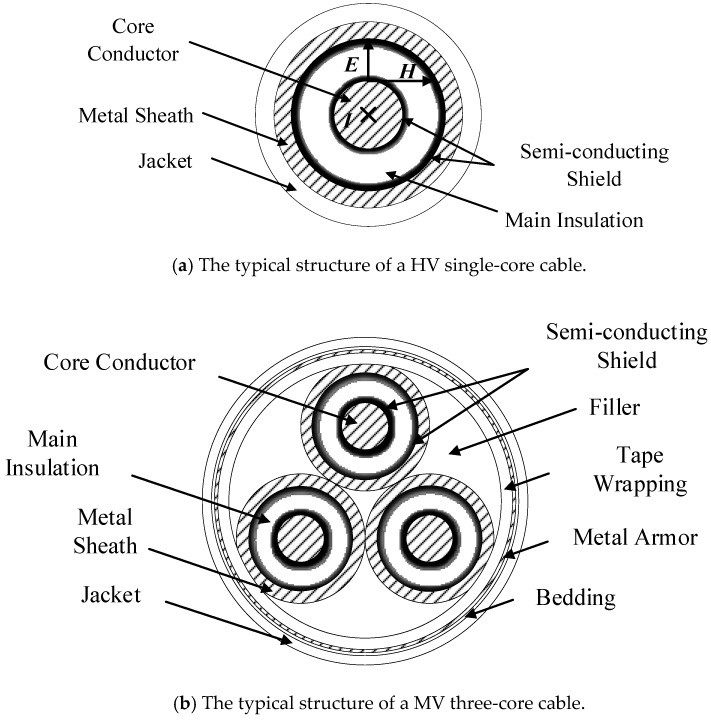
Schematics of typical power cable structures.

**Figure 3 sensors-19-02083-f003:**
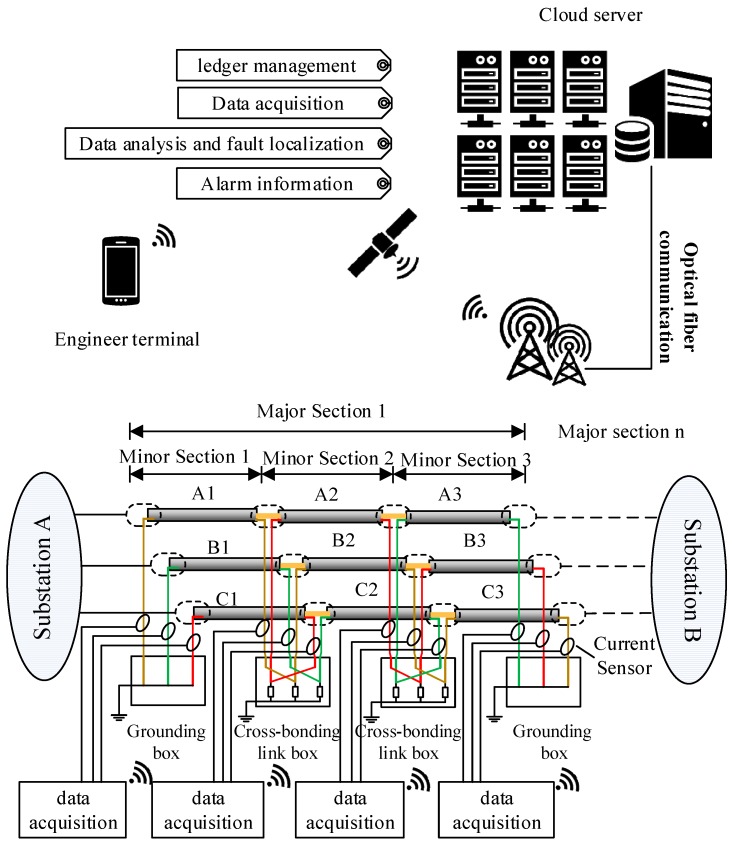
The online sheath currents monitoring system.

**Figure 4 sensors-19-02083-f004:**
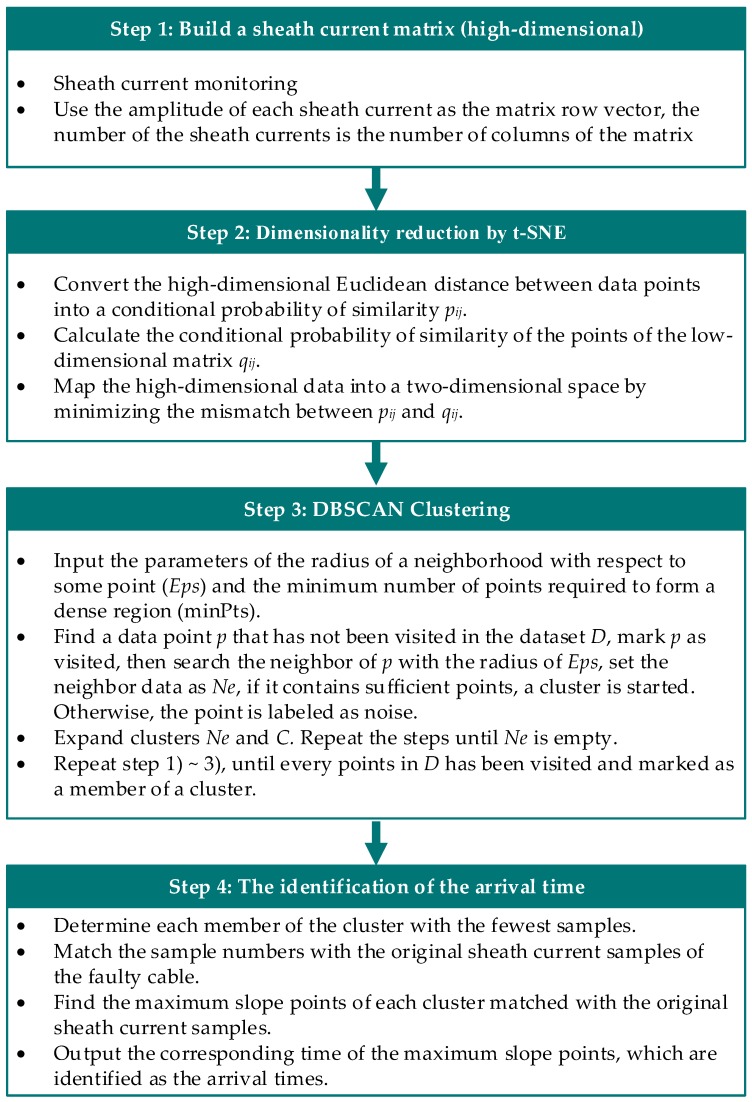
The process of identification of the arrival times at the two ends of the line.

**Figure 5 sensors-19-02083-f005:**
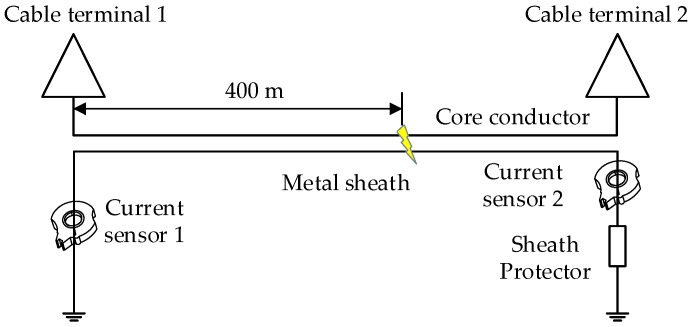
The configuration of a single-point bonding HV cable system.

**Figure 6 sensors-19-02083-f006:**
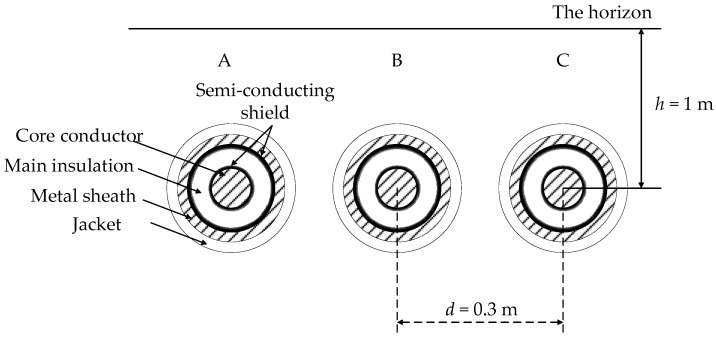
The configuration of a horizontal laid HV (high voltage) cable system.

**Figure 7 sensors-19-02083-f007:**
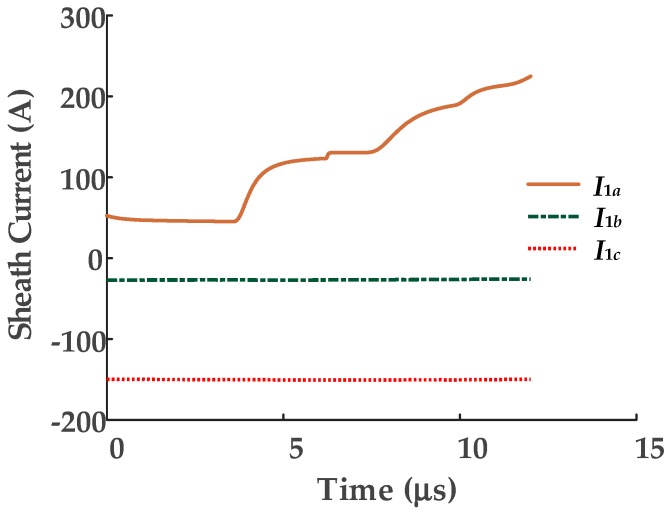
The sheath current wave of the faulty phase at terminal 1 of the cable. (*I*_1*a*_, *I*_1*b*_, *I*_1*c*_ respectively represent the sheath currents recorded by the current sensors on phases A, B, and C).

**Figure 8 sensors-19-02083-f008:**
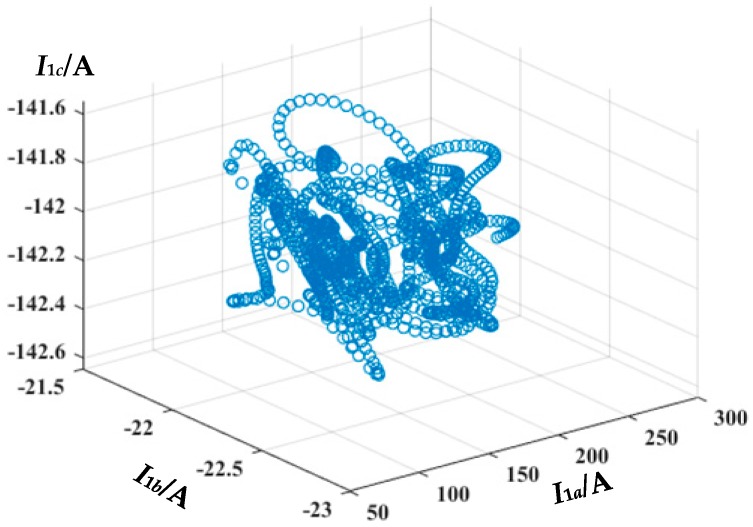
The 3D map of the sheath current matrix *U*.

**Figure 9 sensors-19-02083-f009:**
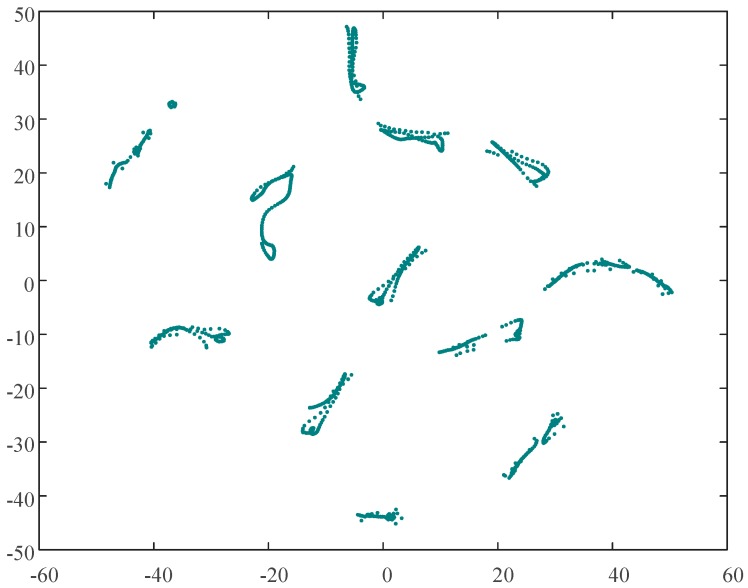
The 2D map of the sheath current matrix *V*.

**Figure 10 sensors-19-02083-f010:**
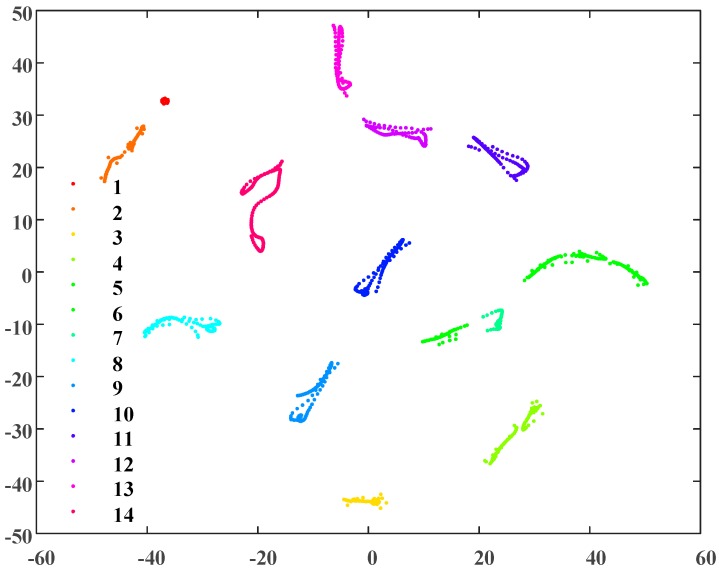
The DBSCAN (density-based spatial clustering of applications with noise) clustering results of the sheath current matrix *V*.

**Figure 11 sensors-19-02083-f011:**
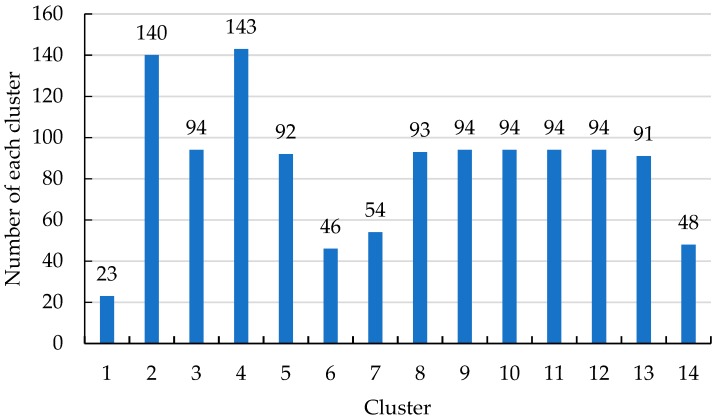
The number of the sheath current samples in each cluster.

**Figure 12 sensors-19-02083-f012:**
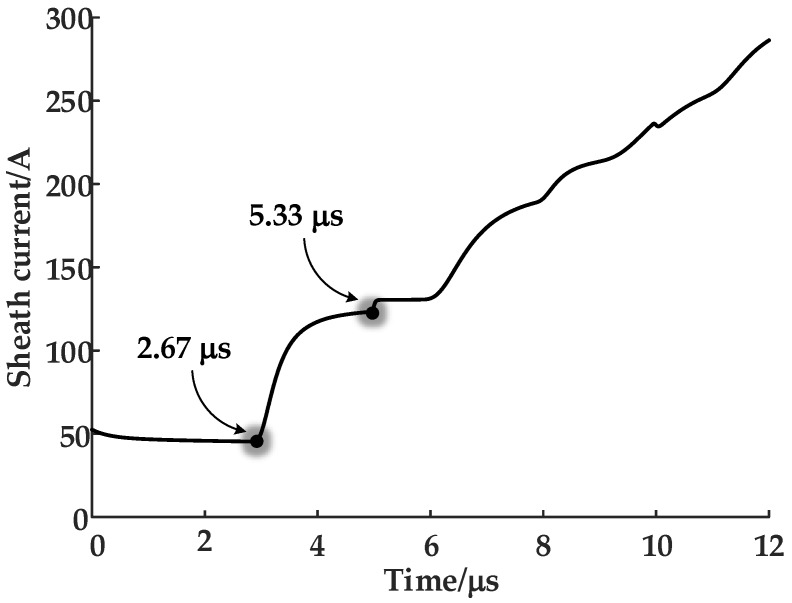
The sheath current wave of the faulty phase at terminal 1 of the cable.

**Figure 13 sensors-19-02083-f013:**
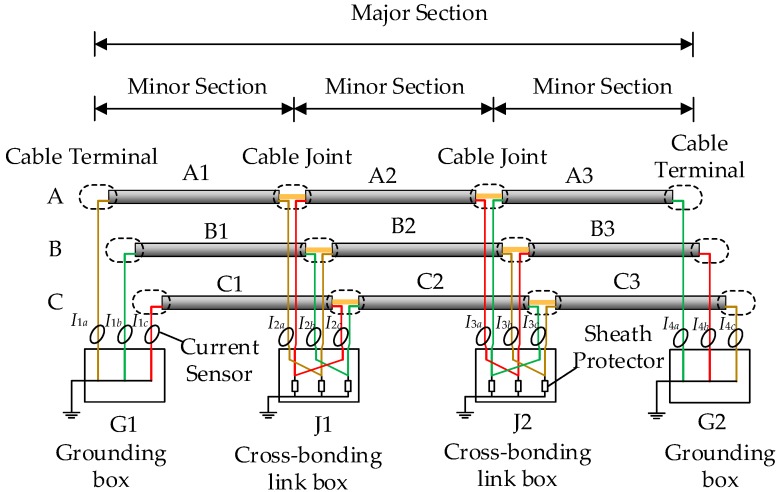
The schematic diagram of the cross-bonded cable circuit. (*I*_1*a*_, *I*_1*b*_, *I*_1*c*_ are, respectively, the sheath currents recorded by the current sensors of phase A, B, C at G1; *I*_2*a*_, *I*_2*b*_, *I*_2*c*_ are, respectively, the sheath currents recorded in phase A, B, C at J1; *I*_3*a*_, *I*_3*b*_, *I*_3*c*_ are, respectively, the sheath currents recorded in phase A, B, C at J2; *I*_4*a*_, *I*_4*b*_, *I*_4*c*_ are, respectively, the sheath currents recorded in phase A, B, C at G2.).

**Figure 14 sensors-19-02083-f014:**
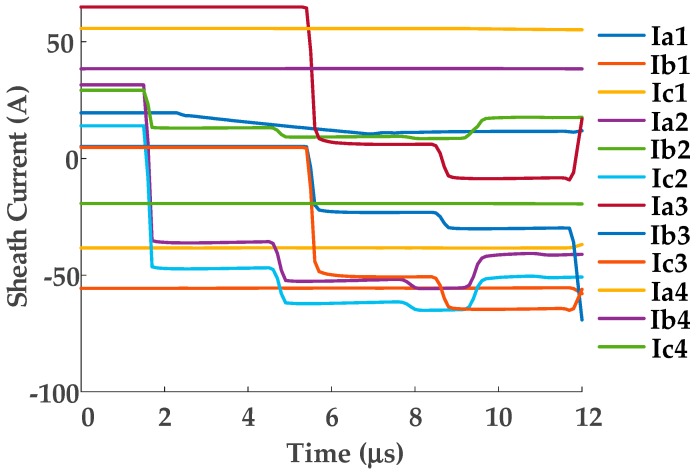
The 12 sheath currents for the cross-bonded HV cable.

**Figure 15 sensors-19-02083-f015:**
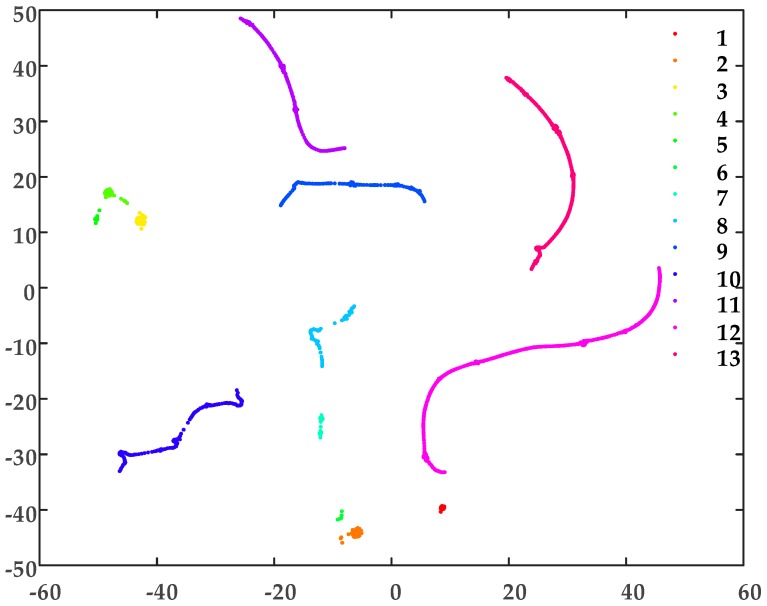
The DBSCAN clustering results for the cross-bonded cable sheath current matrix.

**Figure 16 sensors-19-02083-f016:**
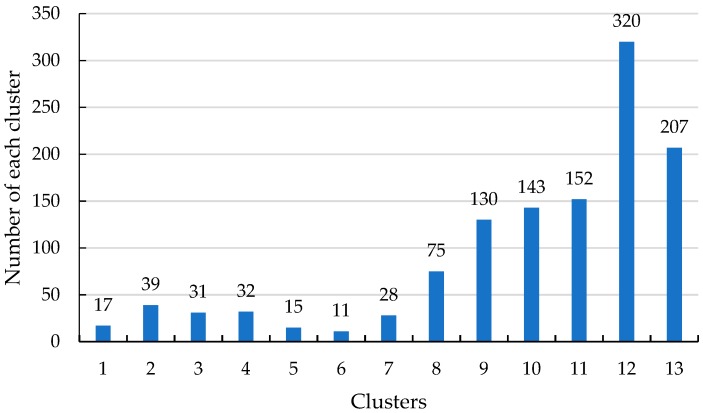
The number of sheath current samples in each cluster.

**Figure 17 sensors-19-02083-f017:**
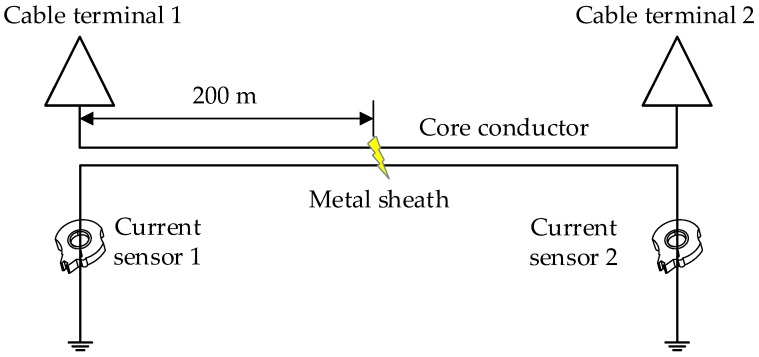
The configuration of a double-end bonded MV cable system: note, ‘core conductor’ has three cores, one for each phase.

**Figure 18 sensors-19-02083-f018:**
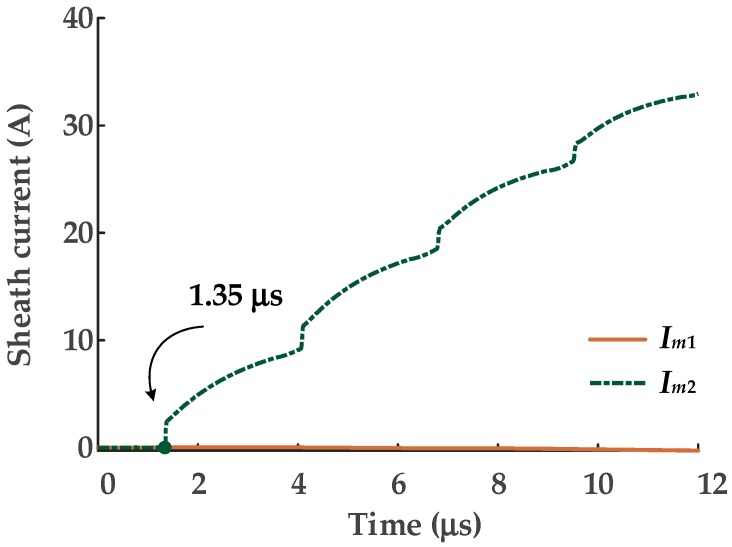
The sheath current waves monitored by the current sensors at the two ends.

**Figure 19 sensors-19-02083-f019:**
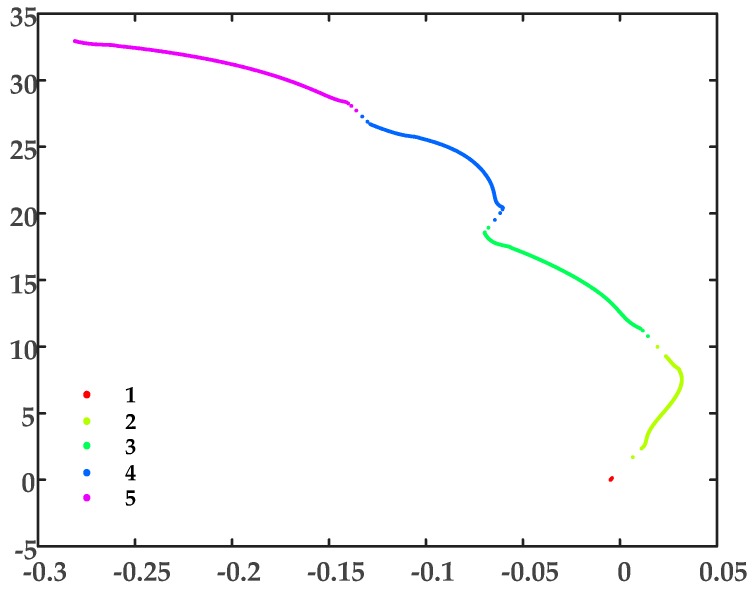
The DBSCAN clustering results for the MV cable sheath current matrix.

**Figure 20 sensors-19-02083-f020:**
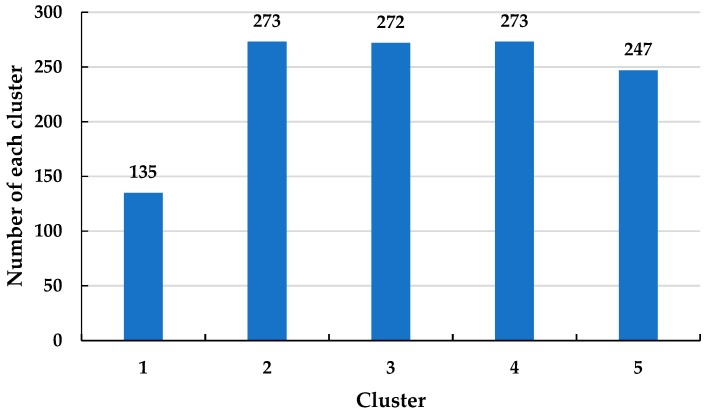
The number of the sheath current samples in each cluster.

**Figure 21 sensors-19-02083-f021:**
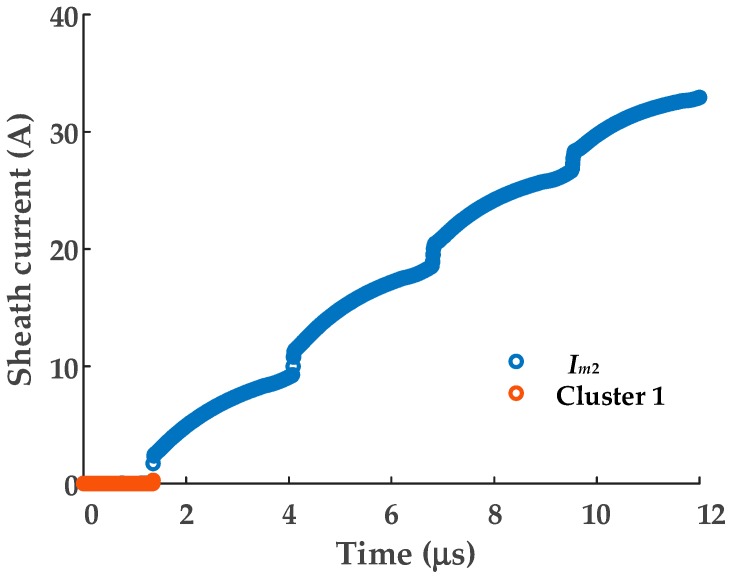
The scatter diagram of the sheath current *I_m_*_2_.

**Figure 22 sensors-19-02083-f022:**
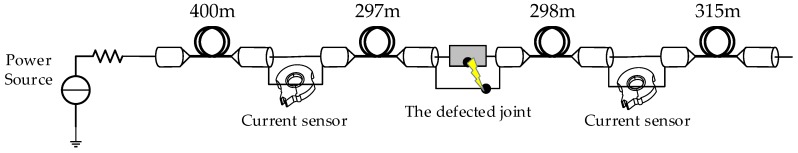
The schematic diagram of the test cable circuit.

**Figure 23 sensors-19-02083-f023:**
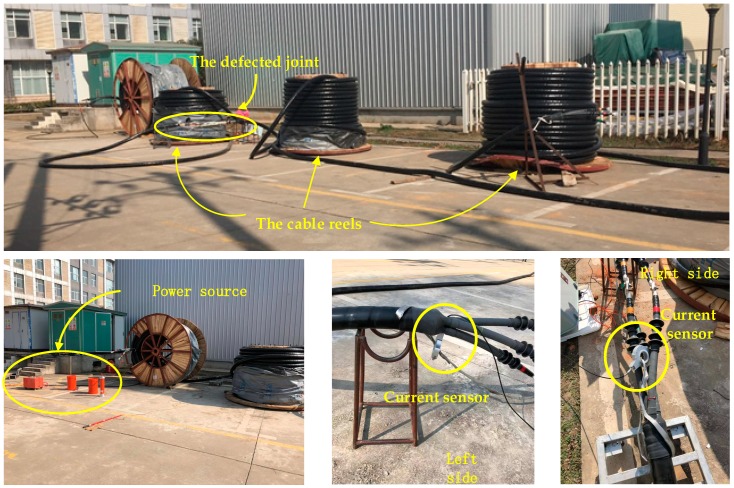
The on-site photos of the test MV (medium voltage) cable circuit.

**Figure 24 sensors-19-02083-f024:**
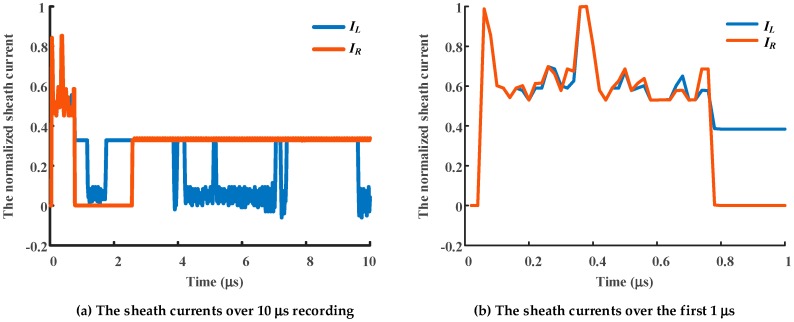
Normalized sheath currents recorded by the current sensors at the positions shown. (*I_L_* is the sheath current recorded by the sensor to the left of the fault, *I_R_* is the sheath current recorded by the sensor to the right of the fault.).

**Figure 25 sensors-19-02083-f025:**
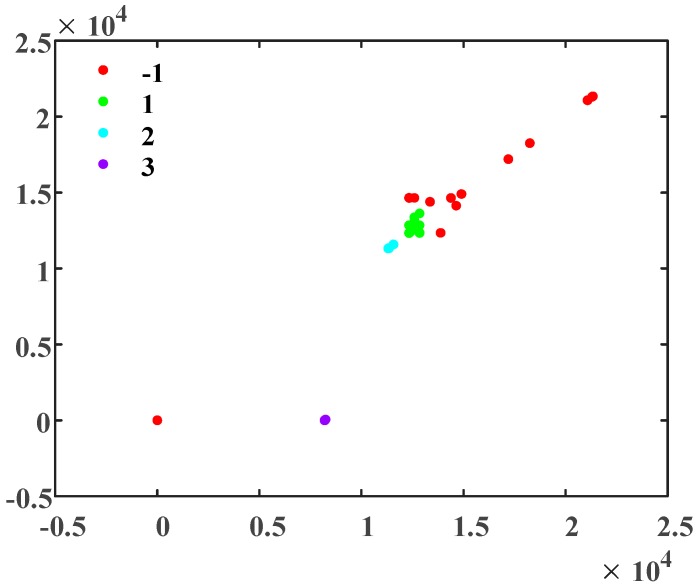
The DBSCAN clustering results for *I_L_* and *I_R_*.

**Figure 26 sensors-19-02083-f026:**
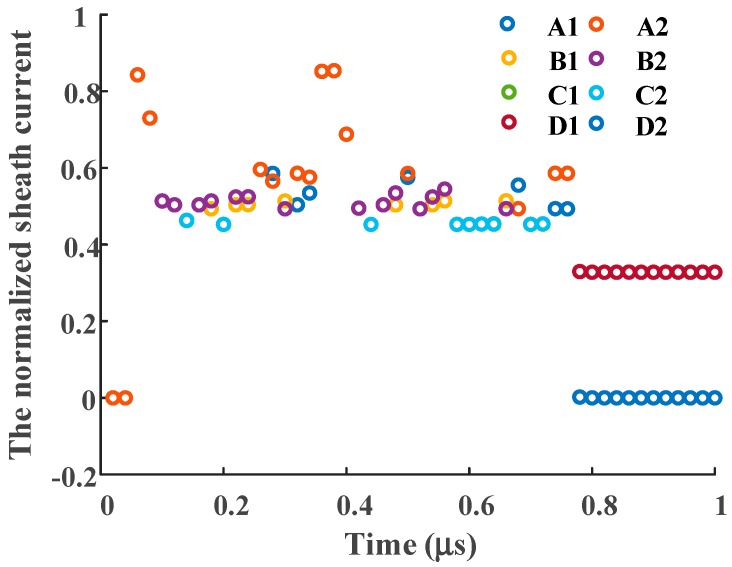
The scatter diagram of the clustering data of *I_L_* and *I_R_*.

**Figure 27 sensors-19-02083-f027:**
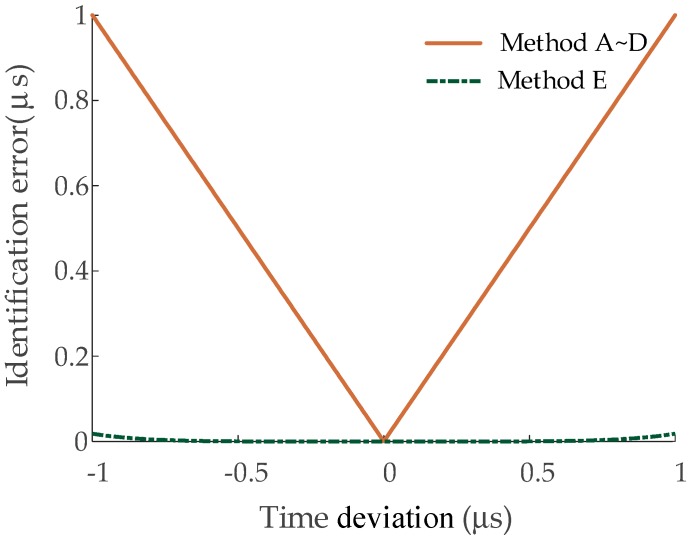
The performances of methods A~E with a time error added to *I*_1*a*_. (The ordinate axis represents the absolute value of the identification error).

**Table 1 sensors-19-02083-t001:** Parameters of cross-sectional structure of the HV cable.

	Structure	Outer Radius/mm
1	Core conductor (Copper)	17.0
2	Inner semi-conductor (Nylon belt)	18.4
3	Main insulation (Ultra-clean XLPE)	34.4
4	Outer semi-conductor (Super-smooth semi-conductive shielding material)	35.4
5	Water-blocking layer (Semi-conductor)	39.4
6	Metal sheath (aluminum)	43.9
7	Jacket (PVC)	48.6

**Table 2 sensors-19-02083-t002:** Parameters of cross-sectional structure of the MV cable.

	Structure	Outer Radius/mm
1	Core conductor (Copper)	10.30
2	Main insulation (including inner and outer semi-conductor)	16.60
3	Metal sheath (Copper tape)	16.75
4	Filler and tape wrapping (semi-conductor)	36.50
5	Metal armor (Steel)	38.50
6	Bedding (Water-blocking PVC)	40.10
7	Jacket (Fire-resistant PVC)	44.90

**Table 3 sensors-19-02083-t003:** The time difference results for the five algorithms under ideal conditions.

	TDI*	2.03 μs	1.35 μs	0.68 μs	0 μs	−0.68 μs	−1.35 μs	−2.03 μs
Method								
A		2.03 μs	1.35 μs	0.68 μs	0 μs	−0.68 μs	−1.35 μs	−2.03 μs
B		2.03 μs	1.35 μs	0.68 μs	0 μs	−0.68 μs	−1.35 μs	−2.03 μs
C		2.03 μs	1.35 μs	0.68 μs	0 μs	−0.68 μs	−1.35 μs	−2.03 μs
D		2.03 μs	1.35 μs	0.68 μs	0 μs	−0.68 μs	−1.35 μs	−2.03 μs
E		2.03 μs	1.35 μs	0.68 μs	0 μs	−0.68 μs	−1.35 μs	−2.03 μs

(*: TDI is short for the time difference ideal).

**Table 4 sensors-19-02083-t004:** The time difference results for the five algorithms with white noise.

	SNR*	50 dB	40 dB	30 dB	20 dB	10 dB	0 dB
Method							
A		−0.68 μs	−0.59 μs	−0.76 μs	−0.54 μs	−0.47 μs	−0.25 μs
B		−0.68 μs	−0.68 μs	−0.63 μs	−0.59 μs	−0.73 μs	−0.37 μs
C		−0.68 μs	−0.68 μs	−0.66 μs	−0.75 μs	−0.75 μs	−0.35 μs
D		−0.68 μs	−0.68 μs	−0.68 μs	−0.67 μs	−0.55 μs	−0.42 μs
E		−0.68 μs	−0.68 μs	−0.68 μs	−0.68 μs	−0.67 μs	−0.53 μs

(*: SNR is short for signal-to-noise ratio).

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
