# Peer review of "A Novel Traveling-Wave-Based Method Improved by Unsupervised Learning for Fault Location of Power Cables via Sheath Current Monitoring"

_sensors, 2019, doi:10.3390/s19092083_

Round 1

Reviewer 1 Report

Authors are advised focus on unsupervised learning as it is the fundamental part of the paper and quite shortly presented in the paper. Do consider improving an overview of methods for unsupervised learning in measurements, that would enable detailed discussion based on the Tables 3 and 4. These methods are briefly introduced in Section 6, however this is not the place. Related to this is also point 5 in Section 7, where with the information provided it is hard to prove, therefore an improvement is needed.

Based on "This paper proposes an autonomous learning mechanism based on the two-terminal traveling wave method, which enables the algorithm to accurately identify the accurate arrival time of a fault traveling wave with multiple monitoring data.", provide what are the conditions that need to be met in regard to the other methods presented in the state of the art and later in results section as a comparison to your method.

Provide also condition how many samples do you need for reliable unsupervised learning? This is not evident at this moment. How does the sample size influences result? I also recommend that facts about measurement deployments are collected together in a new sub-section.

Make sure that the distinction between HV and MV is clear (see point 3 in Section 7). Provide adequate information also in the introduction as well as conclusion of the paper.

Figures are missing detailed information what is on the axes (see for example Figure 24, as well as others), what are the units used, what are we observing. Also explain what are individual dots in the figure, for example Figure 8?

Based on Figure 9 you provide: "Distribution concentration is related to the difference in arrival time at the terminals.". Explain what do you mean and how this related to Figure 9.

How can you confirm that " As shown in Figure 20, the samples of cluster 1 are almost coincident with the values of the first traveling waveform; the amplitude of these samples are closest to the corresponding samples of Im1. The arrival time can be identified as 1.35 ms by the same method, which is also consistent with arrival time of the first traveling wave." Is this is sufficient proof? 

Overall, paper present an interesting approach, however to be published authors need to organise the setup and measurements more systematically. As the work is related to unsupervised learning algorithm detailed state of the art is needed in order to be able to understand the results related to the work of others. 

Reviewer 2 Report

1.-Currents in the cable shields

Why do you measure the current of the sheath? And why not the current in the cables?

Do you have better results?

What is the advantage of metering the currents in the shields?

2.- Step 1

Do you measure the instantaneous values or the rms values?

In the Figure 6 are presented the instantaneous values?

In the healthy phases the current is so flat?

3.- Record of the faults

It would be convenient to include a record of the currents in the three cable for a better understanding of the example presented in the subsection 4.1, 5.1 and 5.3.

4.-Sampling rate.

It is realistic to use a sampling frequency of 100 MHz in a protection relay?

Do you think that this method could be implemented?

5.- Starting of the record to build the current matrix

How is determined the beginning of the 12 µs record?

6.- Figures 6 and 11

Could you please increase the time to show the currents in the sheaths just before the ground fault took place?

7.- Calculation of the fault distance

According to the information the fault distance is calculated measuring the time from the beginning of the fault to the slope changes.

2.67e-6  x  1.48 e 8 =  395.16 meter

If you confirm this, why it is necessary Step 2 and Step 3 to find the maximum slope points.

7.- Cross bounded discontinuities in the link boxes

In the figure 12 is presented a cross bounded high voltage cable.

There are cable shields discontinuities in the link boxes.

It is necessary to measure the sheath currents in all the link-boxes (12 sensors) ?

It is not possible to measure only in one end of the cable (3 sensors) ?

8.- Advantages and disadvantage of the method

According to the comparison of different method, your method is more accurate than the other analysed method. However, is a very complex and expensive (12 sensors in a three section cross bounded cable)?

Could you quantify the improvements in error distances from the real fault location?

What are the disadvantages of your method?

Round 2

Reviewer 1 Report

Paper was significantly improved and can be accepted, therefore I have no further comments.

Only one suggestion remains, it is unusual that discussion follows the conclusion - authors are advised to put the discussion before the conclusion.

Author Response

The authors are grateful for the reviewer’s constructive comments. The discussion has been moved before the conclusion.
